# Perceptions of General Attitudes towards Older Adults in Society: Is There a Link between Perceived Life Satisfaction, Self-Compassion, and Health-Related Quality of Life?

**DOI:** 10.3390/ijerph20043011

**Published:** 2023-02-09

**Authors:** Anna Sofia Bratt, Cecilia Fagerström

**Affiliations:** 1Faculty of Health and Life Sciences, Department of Psychology, Linnaeus University, 35195 Växjö, Sweden; 2Faculty of Health and Life Sciences, Linnaeus University, 39182 Kalmar, Sweden; 3Department of Research, Region Kalmar County, 39236 Kalmar, Sweden

**Keywords:** aging, aged, attitude, stereotyping, quality of life, psychological well-being

## Abstract

Negative attitudes towards aging are common in society. However, few studies have investigated how older adults perceive this phenomenon. This study investigated (a) how older adults in Sweden perceive general attitudes towards the older population and whether negative perceptions are associated with low life satisfaction, self-compassion, and health-related quality of life (HRQL), and (b) whether perceived attitudes predict life satisfaction when controlling for HRQL, self-compassion, and age. The sample comprised 698 randomly selected participants, aged 66–102 years, from the Blekinge part of the Swedish National Study on Ageing and Care. The results showed that 25.7% of the participants held negative attitudes towards older adults and reported lower life satisfaction and HRQL. Self-compassion was related to higher life satisfaction, perceived positive attitudes, and better mental HRQL. Overall, perceived attitudes, HRQL, self-compassion, and age predicted 44% of the participants’ life satisfaction. Understanding the factors that influence older adults’ life satisfaction is crucial, as health-related losses might reduce the opportunity for a successful life. Our study makes an important contribution to the field, showing that perceived attitudes explained 1.2% of the variance of life satisfaction, whereas mental and physical HRQL accounted for 18% of life satisfaction.

## 1. Introduction

The world population’s age structure is changing, with the number of people aged 65 and above expected to double in 2050 to over 1.5 billion [1]. In 2021, the share of the Swedish population aged 65 years or over was about 20.3% (or 2.1 million people) [2]. By 2070, the share is anticipated to reach 25% of the population [3]. A high proportion of older people places an increased strain on the welfare system, especially the health care and the social security system [4]. When financial resources are limited, such as in times of economic crises, structural ageism and negative attitudes towards older adults seem to increase among health care professionals [5]. When health care professionals have limited time, the tendency to stereotype older patients and not give them necessary care increases [5]. A report from the Swedish National Board of Health and Welfare showed that older people, when seeking mental health care, received pharmacological treatment, such as benzodiazepines, instead of psychological [6]. The use of potentially inappropriate medications, that is (medication which has greater potential risk than potential benefit), is a worldwide negative phenomenon creating great health risks for older people, but also unnecessary costs for the health care sector [4].

As they age, older adults face several challenges, including stigmatization and negative attitudes that are common in society, for example, in the media [7]. Stigmatization of older adults can operate at both a structural (i.e., institutions; health care) and individual level [5], becoming a public health issue with negative effects on the health and well-being of older adults [8]. Ageism, or negative attitudes towards older adults, including those held by older adults themselves, is shaped by prejudice, stereotypes, and discrimination [9]. Negative stigmatization because of age can lead to experiences of being excluded or cast in a subordinate role, which in turn decreases psychological and physical well-being [8,10]. Attitudes can be defined as thoughts or beliefs with an evaluative component [11]. How one perceives older adults can differ by age, where individuals who are 80 years and above) face more negative attitudes, such as being perceived with pity or as a burden, compared with individuals below 70 years of age [12]. Attitudes towards one’s personal experience of aging have been associated with well-being, health, and self-compassion [13,14]. Research reveals that individuals who have a positive view of aging, both their own and in general, perceive less discriminatory behaviors in others [12,15]. There is a lack of studies on how older adults in Sweden perceive general attitudes towards the older population and whether there is an association with their own aging process.

The extent to which individuals hold a positive image of themselves and the aging process is part of the concept of life satisfaction [16]. Knowledge of the indicators of older adults’ life satisfaction is essential, because health-related losses might reduce opportunities for those at an advanced age to live a successful life. Internal and external factors that are well-known indicators of low life satisfaction include age—where the oldest individuals reported lower life satisfaction than those aged 70–79 years—negative perceived health [17], personality, and financial and social resources [18]. Self-compassion is one aspect associated with better perceived life satisfaction [13,19] and refers to the ability to take care of and support oneself in moments of suffering [20,21].

Self-compassion has been described as psychological resilience in relation to aging, helping older adults maintain well-being despite challenges such as physical health problems [22]. Older adults who are high in self-compassion seem to have a more accepting attitude towards negative age-related events, such as physical and mental limitations, than those who are low in self-compassion [23]. Increased self-compassion has also been found to strengthen the ability to balance negative emotions related to threatening social circumstances [24]. It could be helpful for older adults facing negative attitudes towards aging from others or themselves [25]. Whether there is a relationship between perceived attitudes towards aging in society and self-compassion remains unknown.

Life satisfaction includes a broader range of an individual’s perceptions of the aging process and self-identity. Health-related quality of life (HRQL) focuses on how physical and mental health affects the day-to-day demands of life and whether the ability to fulfill needs and desires is constrained by a person’s health [26]. Self-rated health has consistently been described as closely related to well-being, life satisfaction [17], and self-compassion [22].

Negative perceptions of age and aging, at societal and individual levels, have adverse effects on older adults’ health and well-being [27]. Existing research reveals that negative attitudes from health care professionals are more prevalent among older women than men [5]. However, results are inconsistent, and some studies show more positive attitudes towards women [5]. There is a lack of studies regarding gender differences in perceived attitudes of older adults. Finally, previous research has shown that age, HRQL, and self-compassion predict life satisfaction in older adults; however, the association between perceived general attitudes towards older adults and their life satisfaction is unknown. To the best of our knowledge, there are no population-based studies on self-compassion in older adults, whereas the existent ones are too few to be generalizable to the older population. As such, we sought to investigate how older adults perceive attitudes towards older adults through a population-based study, taking also into account whether such attitudes can predict life satisfaction.

### Aims

The aim of the current study was fourfold:Investigate how older adults in Sweden perceive general attitudes towards older adultsExplore whether there are gender differences in perceived general attitudesExamine whether older adults who perceive those attitudes as negative differ in levels of life satisfaction, self-compassion, and health-related quality of life (HRQL)Explore whether perceived attitudes can predict life satisfaction when controlling for HRQL, self-compassion, and age

## 2. Materials and Methods

### 2.1. Study Design

This was a randomized, cross-sectional cohort study focused on the Blekinge region, one of the four research areas from the Swedish National Study on Ageing and Care (SNAC), which is a large, national, longitudinal study. The SNAC has been collecting data since 2001, using a randomized selected panel of older adults from 10 age cohorts (66, 72, 78, 81, 84, 87, 90, 93, 96, and 99 years) as its population sample. For more information on SNAC, see Lagergren et al. [28]. The Blekinge region (SNAC-B) encompasses only the Karlskrona municipality, situated in southeastern Sweden, with approximately 66,800 inhabitants from both urban and rural areas. Karlskrona’s population resembles other subpopulations in Sweden in terms of age, sex distribution, and functional ability. The survey data for the present study were collected in the fourth wave between 2013 and 2015. The Regional Ethics Review Board of Lund (LU 128-00, LU 604-00) ethically approved the study.

### 2.2. Participants and Procedure

The sample comprised 698 randomly selected participants aged 66–102 years, with a 73% response rate. Random sampling was implemented in order to select a representative sample from the larger population of Sweden. In this way, we ensured that we could make sound generalizations about the population and minimize any bias. Table 1 shows the socio-demographic characteristics of this sample. An invitation to participate in the study was mailed to potential participants on two occasions. The research team invited those who did not respond to the letter via telephone. Data were collected at the research center, if possible, and otherwise at the respondents’ homes. When necessary, the respondents were offered help in completing the questionnaire. Of the non-respondents (*n* = 254), 128 died before the start of the 2013–2015 study wave, 25 had impaired health, 23 had moved from the area, 15 had not been contacted, and 63 had other reasons.

### 2.3. Measures

#### 2.3.1. Sociodemographic Characteristics

Self-reported single items concerning age, sex, marital status, and education were taken from the SNAC-B database. 

#### 2.3.2. Society’s Attitudes towards Older Adults

One question was asked about attitudes: “What do you think about the attitudes towards older adults in today’s society, for example, in the media?” The responses ranged from 1 (highly negative) to 5 (highly positive). Three groups were created: (1) negative (rather or very negative), (2) neutral (neither negative nor positive), and (3) positive (rather positive or very positive) attitudes. This question is referred to as “perceived attitudes” in the forthcoming text.

#### 2.3.3. Life Satisfaction

Life satisfaction was measured using Neugarten’s Life Satisfaction Index (LSI-A [16]). The LSI-A is used worldwide and includes the extent to which individuals hold positive images of themselves and the aging process. Different versions of the LSI-A have been developed (for more discussion about the psychometric properties, see Fagerström et al. [29]; Helmes et al. [30]; and Liang [31]). The long version of the LSI-A, still the most used, has been described as valid and reliable [32]. Adequate reliability was reported, with an average of 0.79 when 34 samples were examined [33]. The 20 questions included in the LSI-A capture not only life satisfaction but also the respondent’s attitude towards the aging process. Twelve items were positively loaded with the responses agree, doubtful, and disagree, and eight were negatively loaded with the same responses. The original LSI-A method used dichotomous scoring, but a trichotomous approach was found to have better internal consistency, because it used more information from the questionnaire [30]. Scores range from 0 to 40, with high scores indicating high life satisfaction. There were five components of life satisfaction: (1) zest versus apathy indicates the degree of pleasure the participant takes from everyday activities; (2) resolution and fortitude measure whether the respondent regards their life as meaningful and accepts the way it has gone; (3) congruence measures agreement between desired and achieved goals; (4) self-concept measures whether the participant has a positive self-image; and (5) mood measures whether the respondent is happy and optimistic. The Cronbach’s alpha for the total LSI-A was 0.83.

#### 2.3.4. Health

Health-related quality of life was measured using the short-form health survey (SF-12) [34]. The SF12 includes two components or dimensions of HRQL: the mental (MCS, “mental HRQL”) and physical (PCS, “physical HRQL”) component summaries, including a total of 12 questions. The responses are summarized as scores for mental and physical HRQL according to an algorithm [35]. Scores ranged from 0 to 100, with higher scores indicating a higher HRQL. Normative scores for the general Swedish population aged ≥75 years were 51.5 (SD = 11.0) for mental HRQL [31] and 40.3 (SD 11.6) for physical HRQL. In older adults, the instrument has been evaluated as valid for measuring HRQL [36]. The SF-12 dimensions have acceptable internal consistency for older age groups [37]. In the present study, Cronbach’s alpha was α = 0.80 and 0.88 for mental HRQL and physical HRQL, respectively.

#### 2.3.5. Self-Compassion

The 12-item Self-Compassion Scale-Short Form (SCS-SF) was used to measure levels of self-compassion [38,39]. Internal consistency and Cronbach’s alpha were ≥0.86. Cronbach’s alpha in the present study was α = 0.67 for the total sample. The SSC-SF items are rated on a 5-point response scale ranging from 1 (almost never) to 5 (almost always), including three different components of self-compassion consisting of four questions each. The first component measures self-kindness (e.g., “I try to be understanding and patient towards those aspects of my personality that I don’t like”) versus self-judgement (e.g., “I’m disapproving and judgmental about my own flaws and inadequacies”), the second measures common humanity (e.g., “I try to see my failings as part of the human condition”) versus isolation (e.g., “When I am feeling down I tend to feel like most other people are probably happier than I am”), and the third component measures mindfulness (e.g., “When something painful happens I try to take a balanced view of the situation”) versus over-identification (“When I fail at something important to me I become consumed by feelings of inadequacy). The scores ranged from 12 to 60, with higher scores indicating higher self-compassion.

### 2.4. Analysis

The chi-squared test was performed to examine differences between attitude-groups and sociodemographic characteristics (Table 1). One-way between-group ANOVAs were then performed with post hoc comparisons using the Tukey honestly significant difference (HSD) test to explore the differences between attitude-groups for age, life satisfaction, physical HRQL, mental HRQL, and self-compassion (Table 1). Additionally, Pearson’s correlations were conducted to investigate the relationship’s strength and direction between the dependent variable (life satisfaction) and the independent variables (Table 2). Furthermore, a standard multiple regression was used to assess if perceived attitudes could predict the variance in life satisfaction when adjusted for self-compassion, age, and physical and mental HRQL (Table 3). Extra analyses were executed to ensure no violation of normality, linearity, multicollinearity, and homoscedasticity assumptions. All analyses were performed using the SPSS statistical package (version 28).

## 3. Results

Table 1 shows the sample descriptions and information regarding perceived attitudes in society. Nearly half of the participants reported that society’s attitudes towards older adults were neither positive nor negative (47.0%). Slightly more participants perceived positive (27.3%) rather than negative attitudes (25.7%). There were no differences between men and women in terms of their perceived attitudes in society.

Moreover, in Table 1, the one-way between-group ANOVA analyses with post-hoc comparisons using the Tukey HSD test showed that the positive-attitude-group had significantly higher life satisfaction than the negative- or neutral-attitude-groups. The negative-attitude-group had significantly lower HRQL than the positive-attitude-group. Furthermore, there were no significant differences in HRQL between the neutral group and the other two groups. No differences in self-compassion and age were found among the three groups (Table 1).

The correlation analysis in Table 2 revealed that all included variables were significantly correlated with life satisfaction. Mental HRQL had the highest correlation with life satisfaction, followed by physical HRQL, self-compassion, age, and perceived attitudes. Perceived positive attitudes were associated with higher levels of life satisfaction. Higher self-compassion was associated with greater life satisfaction, perceived positive social attitudes, and better mental and physical health.

A standard multiple regression analysis, in Table 3, was conducted to investigate if perceived attitudes could predict the variance in life satisfaction when adjusting for physical and mental HRQL, self-compassion, and age. The result revealed that the included variables significantly predicted life satisfaction: F(5, 503) = 77.48, *p* < 0.01, R^2^ = 43.7. Mental HRQL made the strongest contribution to life satisfaction, followed by physical HRQL, self-compassion, age, and perceived attitude. Part correlation coefficients, which indicate the unique contribution of independent variables, showed that mental HRQL contributed 13.7% to the total R-square; physical HRQL, 4.3%; self-compassion, 2.8%. years, 1.8%; and perceived attitudes, 1.2%.

## 4. Discussion

To our knowledge, this is the first study to examine differences in life satisfaction, health-related quality of life, and self-compassion among older adults who perceive general attitudes towards aging to be negative, neutral, or positive. About 25% of participants perceived attitudes in society as negative, and this group was less satisfied with life and had lower HRQL (both physical and mental) than the other groups. Although self-compassion did not differ between attitude groups, correlation analyses revealed that higher self-compassion was related to higher life satisfaction, perceived positive attitudes, and better mental HRQL. There were no differences between women and men regarding their perceived attitudes. Overall, perceived attitudes, HRQL, self-compassion, and age predicted a substantial (44%) proportion of life satisfaction. The mental and physical HRQL accounted for almost half of the proportion (18%), revealing high importance for life satisfaction at an old age.

Highlighting the stereotypes of older adults is relevant when discussing the stressors related to subjective well-being at an old age. Although life satisfaction is frequently used to measure the subjective well-being of older adults, the literature on predictors of life satisfaction seldom raises the importance of attitudes in the society. Aside from health-related factors, several predictors of life satisfaction have been previously presented in the literature: age, morbidity, impaired functional ability [32], personality, self-esteem [18], and financial and social resources. In this context, the finding that perceived attitudes explained 1.2% of the variance in life satisfaction presents a significant contribution to the existing literature on how older adults feel in old age. A previous study found that about 60% of older adults (80+) felt that they were rather not needed or not at all needed by society, and approximately 14% of them felt rather or strongly treated as a burden by society [40]. Though the feeling of being treated as a burden was highly predicted by health-related variables, their study did not cover the subjective well-being dimension.

In our study, we found no differences in perceived attitudes between women and men. One study found that both older men and women are largely absent in Swedish media such as newscasts, magazines, or television [7]. According to that study, media contributes to ageism, as it fails to depict older adults’ relationships or sexuality. The media content is also gendered, showing stereotypical female versus male gender codes. However, a more positive stereotype of ageing has been emerging in the media, with older women giving advice on early pension arrangements or managing health problems [7]. Research on ageism or attitudes towards older men and women has produced inconclusive results. Some studies show that there are no gender differences, while others have revealed more negative attitudes towards older women [5]. However, there are also studies showing that older women tend to be perceived more positively, particularly by health care professionals [41].

Although the reason that those who experienced negative attitudes in society also had the lowest HRQL is unknown, one explanation is that those with mental and physical problems experience more negative attitudes towards themselves and other older adults in the same situation. Older adults are described as being less well-served by both mental and physical health services than younger adults [42]. Raising awareness of negative attitudes and treatment of older adults, and consciously using this knowledge in the care of older fail persons might support an increased quality of life. The topic of stereotypes highlights the need to have a normative discussion about subjective well-being and finding ways to improve it [40].

Altogether, the older adults in the present study had higher levels of self-compassion than the younger adults (M = 41.59 vs. M = 36.00 [38]), and, as found in earlier studies [43,44], self-compassion was positively related to and predicted about 3% of total life satisfaction. Although self-compassion did not differ between the attitude groups, correlation analyses revealed that higher self-compassion was associated with perceived positive attitudes. Having a positive, caring, and compassionate attitude has been hypothesized to buffer against negative attitudes in society [25]. The findings highlight the need for further studies to deepen the knowledge concerning buffering explanatory factors of health and subjective well-being of older adults. 

### Strengths and Limitations

Strengths of this study include its population-based, randomly selected sample, the reasonably even gender distribution of 56% women and 44% men, and the broad age cohort. The instruments used are well-known and have shown good reliability and validity in other studies [33,37,38].

This study also has some limitations. As the present study had a cross-sectional design, causal relationships between perceived attitudes, life satisfaction, HRQL, and self-compassion cannot be explained. Further experimental studies are required to determine whether a causal link can be established. Another limitation is that perceived attitudes were measured using only questions about general social attitudes towards older adults, not those felt as directed to the individual per se. Moreover, the participants lived in only one part of Sweden and may not mirror the circumstances of the Swedish older population. Whether high levels of self-compassion buffer against negative attitudes could not be answered in this study. Further studies are needed to investigate whether there are differences in attitudes between smaller urban areas and larger cities, as well as whether attitudes vary in different age groups.

## 5. Conclusions

Knowing what indicates older adults’ life satisfaction is essential, as losses related to declining health in advanced age might limit opportunities for a successful life. Therefore, the finding that perceived attitudes explained 1.2% of the variance in life satisfaction presents an important contribution to the field. Most older adults in this study reported that the perception of society’s attitudes towards the older population is neutral. There were no differences in perceived attitudes between women and men. About a quarter of the sample population perceived negative attitudes, were less satisfied with life, and had a lower mental and physical HRQL. This result needs to be given more attention in the research and care of older adults. Self-compassion was related to greater life satisfaction, perceived positive attitudes, and better mental HRQL. Overall, perceived attitudes, HRQL, self-compassion, and age predicted a substantial proportion of life satisfaction (44%). The mental and physical HRQL accounted for almost half the proportion (18%), revealing their high importance for life satisfaction in one’s old age. Moreover, the impact of individuals’ low levels of self-compassion on health and well-being needs further exploration.

## Figures and Tables

**Table 1 ijerph-20-03011-t001:** Sociodemographic characteristics and mean differences by attitudes-groups.

Attitudes-Groups (*n* = 560)								
Variables (*n*)	*n* (%) Negative *n* = 144 (25.7)		*n* (%) Neutral *n* = 263 (47.0)		*n* (%) Positive *n* = 153 (27.3)		*p*-Value	
Gender (698)	Women	Men	Women	Men	Women	Men	ns ^a^	
	90 (28.8)	54 (21.8)	147 (47.1)	116 (46.8)	75 (24.0)	78 (31.5)		
Marital status (665)							ns ^a^	
Married	98 (24.5)		190 (47.0)		116 (28.7)			
Widowed	21 (28.4)		35 (47.3)		18 (24.3)			
Unmarried	8 (30.8)		11 (42.3)		7 (26.9)			
Divorced	14 (32.6)		20 (46.5)		9 (20.9)			
Education (664)							ns ^a^	
Primary school	43 (23.9)		76 (42.2)		61 (33.9)			
Secondary school	22 (27.2)		42 (51.9)		17 (21.0)			
Upper, secondary	20 (26.7)		29 (38.7)		26 (34.7)			
College or higher	20 (26.7)		29 (38.7)		26 (34.7)			
	Negative *n* = 144		Neutral *n* = 263		Positive *n* = 153		Total group (*n* = 560)	*p*-value
	M (SD)		M (SD)		M (SD)		M (SD)	
Age	75.4 (8.09)		74.9 (8.23)		74.8 (8.20)		75.0 (8.18)	ns ^b^
Life satisfaction	26.53 (6.97)		27.60 (7.02)		30.47 (5.92)		28.1 (6.9)	*** ^b^
Physical HRQL	43.28 (11.39)		45.22 (11.05)		46.67 (10.56)		45.14 (11.06)	** ^b^
Mental HRQL	52.91 (8.69)		53.76 (8.21)		55.71 (7.14)		54.07 (8.12)	** ^b^
Self-compassion	40.65 (5.82)		41.87 (5.3)		41.97 (5.02)		41.59 (5.38)	ns ^b^

Note: ^a^ Chi-squared test. ^b^ One-way between-group ANOVA with Tukey HSD post-hoc tests ns= not significant ** *p* < 0.01, *** *p* < 0.001. Missing: marital status *n* = 33, education *n* = 34, attitudes, 138; physical and mental HRQL, 253; attitudes, 138; self-compassion, 104.

**Table 2 ijerph-20-03011-t002:** Pearson Product–Moment correlations between dependent and independent variables.

Variable	1	2	3	4	5
1. Life satisfaction					
2. Attitudes	0.209 **				
3. Physical HRQL	0.418 **	0.111 **			
4. Mental HRQL	0.536 **	0.126 **	0.239 **		
5. Self-compassion	0.312 **	0.087 *	0.079	0.268 **	
6. Age	−0.311 **	−0.031	−0.419 **	−0.147 **	−0.013

Note: The number of cases varies from 509 to 698. * *p* < 0.05 ** *p* < 0.01. Coding: attitudes: negative = 1, neutral = 2, positive = 3.

**Table 3 ijerph-20-03011-t003:** Multiple regression coefficients for independent variables on life satisfaction.

	Unstandardized Coefficients		Standardized Coefficients	95% Confidence Intervals		*p*-Value	Unique Explained Variance by Variable of Total R Square in Percentage
Variable	B	Std. Error	Beta	Lower bound	Upper bound		
(Constant)	0.170	3.401		−6.51	6.852		
Attitudes	1.069	0.322	0.113	0.437	1.701	***	1.2
Physical HRQL	0.147	0.024	0.235	0.100	0.193	***	4.3
Mental HRQL	0.336	0.030	0.396	0.276	0.396	***	13.7
Self-compassion	0.228	0.045	0.175	0.139	0.317	***	2.8
Age	−0.111	0.028	−0.149	−0.166	−0.057	***	1.8

Note: R^2^ = 0.437 *** *p* ≤ 0.001.

## Data Availability

All data generated or analyzed during this study are included in this published article.

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
