# Peer review of "Perceptions of General Attitudes towards Older Adults in Society: Is There a Link between Perceived Life Satisfaction, Self-Compassion, and Health-Related Quality of Life?"

_ijerph, 2023, doi:10.3390/ijerph20043011_

Round 1

Reviewer 1 Report

Dear Authors

"Perceptions of General Attitudes Towards Older People in Society: Is There a Link Between Perceived Life Satisfaction, Self-Compassion, And Health-Related Quality of Life?" - The introduction was well-written.

Major comment:

- In table 1, 153 people showed positive attitude and 144 people showed negative attitude. But in table 2, you mentioned that 144 people showed positive attitude and 153 people showed negative attitude. How is that possible?

- Since you changed the values in table 2, the inferences based on that "Perceived positive attitudes were associated with higher levels of life satisfaction. Higher self-compassion was associated with higher life satisfaction, perceived positive social attitudes, and better mental and physical health", the discussion and conclusion based on that result, everything needs to be reframed.

- Limited discussion was written. Discussion should be elaborate comparing with other related studies.

Minor comment:

- Mention the study design in the methodology

- Place Zero before the decimal throughout the manuscript to avoid misreading

Author Response

Thank you for your valuable comments. I have revised the manuscript according to your feedback. Revisions are marked in yellow.

  1. Comment: In table 1, 153 people showed positive attitude and 144 people showed negative attitude. But in table 2, you mentioned that 144 people showed positive attitude and 153 people showed negative attitude. How is that possible?

Authors answer:This has been corrected. Sorry for this mistake.

  1. Comment: Since you changed the values in table 2, the inferences based on that "Perceived positive attitudes were associated with higher levels of life satisfaction. Higher self-compassion was associated with higher life satisfaction, perceived positive social attitudes, and better mental and physical health", the discussion and conclusion based on that result, everything needs to be reframed.

Authors answer:This was somewhat unclear, the text you are referring to is related to Table 2, the correlation analyses. I have clarified this and marked it yellow in the text. I also changed so I now have both descriptive statistic and the Anova analysis in Table 1, for clarification.

  1. Comment: Limited discussion was written. Discussion should be elaborate comparing with other related studies. Authors answer:More information to the discussion section has been added.
  2. Mention the study design in the methodology. Authors answer: A study design section was added, in 2.1.
  3. Place Zero before the decimal throughout the manuscript to avoid misreading. Authors answer: According to APA manual, 7th edition, zero should only be placed before the decimal when when the statistic can exceed 1. This was checked in the manuscript.

Reviewer 2 Report

The quality of life and life satisfaction of the elderly are attractive fields for readers and this study has entered this field well. This study has some shortcomings, which are described as follows:

The conclusion based on the findings of the study should be added to the abstract.

The keywords should be modified based on the MeSH.

In the introduction, it is expected to provide statistics on the aging process in the research community.

It is also better to mention the health challenges and problems of the elderly

For this purpose, you can use the following study:

·         A Systematic Review of Potentially Inappropriate Medications Use and Related Costs Among the Elderly

In general, in the introduction, you can express the necessity of the study with stronger reasons.

The work method should be written in a more structured way

First, state the type of study. Then write about the research environment, the research community, and then about the sample and how to select it.

Explain about SNAC. How many people did SNAC examine in your study area? Was it a census or sampling?

Regarding the sample size, explain how you arrived at this number?

Why did you do random sampling?

The relationship between variables may change due to the effect of confounders. What have you done to control confounders?

However, the strong point of the methods is the detailed description of the tools.

In the results, table 1 needs major corrections. Please show the attitudes-groups based on the sociodemographic characteristics such as sex, age and etc.

In the discussion, comprehensively discuss all findings and compare them with previous studies

Author Response

Thank you for your valuable comments. I have revised the manuscript according to your feedback. Revisions are marked in yellow.

  1. The conclusion based on the findings of the study should be added to the abstract. Authors answer: The conclusions have been added in the abstract.
  2. The keywords should be modified based on the MeSH. Authors answer:This has been changed.
  3. In the introduction, it is expected to provide statistics on the aging process in the research community. Authors answer:This has been added.
  4. Health challenges and problems of the elderly and more information in the introduction, for the necessity of the study with stronger reasons. Authors answer:This has been added.
  5. The work method should be written in a more structured way. First, state the type of study. Then write about the research environment, the research community, and then about the sample and how to select it. Authors answer: A study design section was added, in 2.1.
  6. Explain about SNAC. How many people did SNAC examine in your study area? Was it a census or sampling? Authors answer: information added in the method part.
  7. Regarding the sample size, explain how you arrived at this number? Authors answer: at the base line the SNAC Blekinge data base included 1402 participants. In the fourth wave 698 persons were left. The reasons of drop outs were that they had deceased, moved to another place or thy do not want to participate in the study any longer.
  8. Why did you do random sampling? The authors answer: we used a random sampling to ensure that results obtained from the sample should approximate what would have been obtained if the entire population had been measured.
  9. The relationship between variables may change due to the effect of confounders. What have you done to control confounders? Author answer: we clarified in the result section that we in the multiple regression analysis controlled for the other variables to see whether the variable perceived attitudes could predict life satisfaction.
  10. In the results, table 1 needs major corrections. Please show the attitudes-groups based on the sociodemographic characteristics such as sex, age and etc. Authors answer: This has been changed.
  11. In the discussion, comprehensively discuss all findings and compare them with previous studies. Authors answer: The results have been discussed more comprehensively in the discussion section.

Round 2

Reviewer 1 Report

Dear Author

In table 1, the mean and SD value for Mental HRQL for total group is missing. 

Author Response

Thanks for noticing the mistake that mean for mental HRQL was missing in table 1. This has been corrected.

Reviewer 2 Report

The authors have made most corrections, but there are a few comments that not adequately addressed yet.

Abstract:

In the conclusion, present your own suggestions based on the findings of the study.

Introduction:

Provide information about aging trends and related statistics in Sweden. What percentage of Sweden's population is elderly.

Also mention the consequences of aging for society and the economic effects and costs of medicines and health interventions for the elderly.

Author Response

Comments:

In the conclusion, present your own suggestions based on the findings of the study. Author respons: This has been added.

Introduction:

Provide information about aging trends and related statistics in Sweden. What percentage of Sweden's population is elderly. This has been added.

Also mention the consequences of aging for society and the economic effects and costs of medicines and health interventions for the elderly. This has been added.
